# CXCL12 Gene Polymorphisms and Serum Levels: Associations with Multiple Sclerosis Prevalence and Clinical Parameters in Lithuania

**DOI:** 10.3390/ijms25179554

**Published:** 2024-09-03

**Authors:** Paulius Valiukevicius, Kriste Kaikaryte, Greta Gedvilaite-Vaicechauskiene, Renata Balnyte, Rasa Liutkeviciene

**Affiliations:** 1Faculty of Medicine, Medical Academy, Lithuanian University of Health Sciences, A. Mickeviciaus 9, 44307 Kaunas, Lithuania; 2Laboratory of Ophthalmology, Neuroscience Institute, Medical Academy, Lithuanian University of Health Sciences, Eiveniu 2, 50161 Kaunas, Lithuania; 3Department of Neurology, Medical Academy, Lithuanian University of Health Sciences, Eiveniu 2, 50161 Kaunas, Lithuania; 4Department of Ophthalmology, Lithuanian University of Health Sciences, Eiveniu 2, 50161 Kaunas, Lithuania

**Keywords:** multiple sclerosis, CXCL12 ELISA, *CXCL12* SNP, chemokine

## Abstract

Our study aimed to investigate the associations between *CXCL12* rs1029153, rs1801157, and rs2297630 single-nucleotide polymorphisms (SNPs), CXCL12 protein levels, MS prevalence, and clinical parameters. This study included 250 individuals diagnosed with MS and 250 sex- and age-matched healthy control individuals from Lithuania. The SNPs were genotyped with real-time PCR-based assays. The CXCL12 protein concentration was evaluated in serum using the ELISA method. Of the studied *CXCL12* SNPs, we found that the rs1801157 CT genotype in the males was associated with 2.3 times reduced MS odds when compared with the CC genotype according to the overdominant and codominant models (*p* = 0.011 and *p* = 0.012, respectively). There was a tendency, which did not reach adjusted statistical significance, for a lower CXCL12 protein concentration in the healthy individuals with the rs1801157 CT genotype (*p* = 0.028). Sensory symptoms were rarer in the women with the rs1801157 TT genotype (*p* = 0.004); however, this genotype was also associated with a shorter MS disease duration (*p* = 0.007). *CXCL12* rs1801157 was associated with reduced odds of MS occurrence in the male individuals. In women, rs1801157 was associated with a sensory symptom prevalence.

## 1. Introduction

Multiple sclerosis (MS) is a chronic inflammatory and demyelinating disease of the central nervous system (CNS), which affects approximately 3 million people worldwide [1]. MS is typically diagnosed between the ages of 20 and 40 [2]. MS imposes a considerable socioeconomic burden since it primarily affects younger individuals, requires lifelong disease-modifying therapy (DMT), and results in significant disability [3]. 

MS results from a complex interplay between genetic and environmental factors. Currently, the most studied demographic risk factors are smoking, vitamin D deficiency, Epstein–Barr virus infection (EBV), and obesity [4]. Women are 2–3 times more likely to develop MS than men, which suggests that hormonal differences also play a significant role [5]. Finally, several HLA genes, such as HLA-DRB115:01 and HLA-DQA101:01, along with non-HLA genes, including NFKB1, IL2Ra, and TYK2, have been associated with an increased risk of MS. In fact, there are more than 200 identified genetic loci associated with MS [6]. A recent study demonstrated that by combining an MS genetic risk score with demographic factors, such as age and sex, it is possible to significantly enhance the prediction of MS among patients presenting with optic neuritis (ON), which is a critical distinction that could guide timely and targeted treatment strategies [7]. 

The pathogenesis of MS is complex and results in CNS inflammation due to immune cell infiltration, cytokine release, and microglia activation [8]. Ultimately, this leads to axonal demyelination, damage, and neurodegeneration, which causes a characteristic wide array of neurological symptoms [9,10]. Soluble signaling proteins, such as cytokines and chemokines, play a significant role in MS pathogenesis. Many non-HLA single-nucleotide polymorphisms (SNPs) associated with MS are located in genes that encode chemokine and cytokine proteins or their receptors, such as IL7R and TNFRSF1A, which have been associated with the attack severity and frequency [11]. These SNPs likely influence the production of bioactive molecules or the downstream signaling of immune cell receptors, which result in an aberrant immune response [12]. 

C-X-C motif chemokine 12 (CXCL12), also known as stromal cell-derived factor 1 (SDF-1), is thought to be important in the pathogenesis of MS. It is a pleiotropic chemokine: it guides neural progenitor cells during embryonic neurogenesis, participates in angiogenesis, and attracts leukocytes during inflammation [13]. In MS patient CNS tissue, CL12 expression is redistributed toward the vessel lumen within active lesions, which promotes autoreactive lymphocyte extravasation. Additionally, tissue CXCL12 levels are correlated with increased leukocyte infiltration and the severity of histological disease [14]. It could also be associated with plasma cell persistence since plasma cells retain the CXCR4 receptor during differentiation. Also, CXCL12 is associated with the astrocyte production of B-cell survival factor [15] and cortical lesions, which confirms the link between B-cell-associated CNS inflammation and adjacent cortical pathology [16]. Such observations could have clinical prognostic value, as cortical pathology has been recognized as a crucial substrate for the progression and irreversible clinical and cognitive disability in MS [17]. Furthermore, the breakdown of CXCL12 by metalloproteinases results in neurotoxic byproducts that contribute to neurodegeneration near MS lesions [15]. Finally, high cerebrospinal fluid (CSF) CXCL12 levels are detected in patients with increased grey matter lesion number and volume, which shows that CXCL12 could be used as a marker of MS progression [18]. 

Our study evaluated three *CXCL12* SNPs: rs1029153, rs1801157, and rs2297630. Only rs1801157, which is in the 3′ untranslated region (UTR) of the gene, was studied in MS patients. rs1029153 and rs2297630 (3′ UTR and intron variants, respectively) have been implicated as possibly significant in other diseases, such as type 2 diabetes, primary pediatric immune thrombocytopenia, and response to hepatitis C treatment [19,20,21], which indicates that these SNPs may have an impact on *CXCL12* expression. Given the significant involvement of CXCL12 in the inflammatory cascade associated with MS, we chose to explore these polymorphisms as potential factors that influence genetic susceptibility and various aspects of the disease.

In this study, we compared the prevalence of specific *CXCL12* gene SNPs (rs1029153, rs1801157, and rs2297630) between MS patients and healthy controls in Lithuania. We also assessed *CXCL12* protein levels in the sera of these groups. Furthermore, we explored the relationships between *CXCL12* SNPs and various clinical parameters of MS, including the age at diagnosis, disease duration, phenotypic variation, affected functional systems, and progression rate. 

## 2. Results

### 2.1. CXCL12 Single-Nucleotide Polymorphism Was Associated with MS Risk

This study included 250 individuals with diagnosed MS and 250 control individuals. The control group was selected to match the age and gender distributions of the MS group. There were no statistically significant differences between the MS and control group gender and age distributions (data are presented in Table 1).

The rs1029153, rs1801157, and rs2297630 genotypes within the MS and control groups were distributed according to the Hardy–Weinberg equilibrium (HWE), with no statistically significant deviations observed. Furthermore, the analysis revealed no statistically significant differences in the distributions of *CXCL12* SNP genotypes between the MS and control groups (Table 2).

After conducting a binary logistic regression analysis of the *CXCL12* genetic models (rs1029153, rs1801157, and rs2297630) in the study groups, no significant associations were found between the examined SNPs and the occurrence of MS (Table 3).

We conducted additional analyses after stratifying the groups by age (≤40 or >40) and gender to determine whether age-specific or gender-specific variations influenced the association between the *CXCL12* SNP genotypes and the MS incidence. 

Age-stratification into two groups (<40 years and ≥40 years) did not result in statistically significant differences between the MS and control group *CXCL12* SNP genotypes or allele distributions. The rs1801157 SNP exhibited a trend toward a higher frequency of C alleles in the MS subjects under 40 years old compared with the controls (*p* = 0.022, Appendix A). However, this result did not achieve statistical significance after an adjustment for multiple comparisons (adjusted *p* > 0.0167). The binary logistic regression did not reach the required statistical significance for all studied *CXCL12* SNPs across the age-stratified groups (*p* > 0.0167) (Appendix A). The gender-stratified analysis of the *CXCL12* SNP distribution also did not reveal any statistically significant differences between the control and MS groups (*p* > 0.0167, Appendix A). There was, however, a tendency in the male group for a higher rs1801157 CC genotype frequency in the MS subjects when compared with the controls (*p* = 0.037). Similarly, the rs1801157 C allele frequency was also higher in the male MS subjects when compared with the controls (*p* = 0.034).

Gender-stratified binary logistic regression demonstrated statistically significant associations for the rs1801157 SNP in males (Table 4). According to the codominant model, the CT genotype reduced the odds of MS by 2.3 times when compared with the CC genotype, with an odds ratio (OR) of 0.43 (95% CI: 0.22–0.83) and *p* = 0.012. In the overdominant model, the CT genotype also reduced the odds of MS by 2.3 times when compared with the combined TT and CC genotypes, with an odds ratio (OR) of 0.43 (95% CI: 0.23–0.83) and *p* = 0.011. Both the codominant and overdominant models fit the data similarly, with AIC values of 229.004 and 229.016, respectively.

### 2.2. CXCL12 Serum Levels in MS Patients and Healthy Subjects

We assessed the serum CXCL12 protein levels in 40 individuals from each group, which were matched for age and gender. For two control group individuals, the CXCL12 protein levels were above the detection range; therefore, these values were excluded from the analysis. The control and MS groups were uniform in their age and gender distributions (Appendix A).

The median (IQR) CXCL12 serum protein concentrations in the control and MS groups were 36.6 (6.8) and 35.4 (10.3), respectively, with a not statistically significant tendency for a higher CXCL12 concentration in the control group (*p* = 0.166). There were no statistically significant differences in the CXCL12 protein concentration when comparing across genders or age groups in both the control and MS subjects (Appendix A). 

To assess whether there was an association between the *CXCL12* (rs1029153, rs1801157, and rs2297630) SNPs and the concentration of the protein encoded by this gene in the serum, an analysis was performed that compared the protein concentrations in the control and MS groups between different SNP genotypes. We did not find any statistically significant differences (Table 5). We also assessed whether there were differences across genotypes in the control or MS groups. There was a tendency for higher CXCL12 protein concentrations among the control group subjects with the rs1801157 C/T genotype when compared with the C/C genotype; however, it did not reach adjusted statistical significance (*p* = 0.028). 

### 2.3. Patients with Multiple Sclerosis Clinical Characteristics and Their Associations with CXCL12 SNP Genotypes

The median (IQR) duration of MS in the MS group was 5 (7) years, with 31 (15) years of age at diagnosis. The EDSS was 3.5 (2.0) and MSSS 5.9 (2.1). The disease phenotypes within the MS group were as follows: the most prevalent was RRMS (90%), followed by PPMS (4.8%), CIS (4%), and SPMS (1.2%). Regarding the presence of symptoms, 87.2% of the group exhibited pyramidal symptoms, 63.6% had sensory symptoms, 58.4% were affected in the brain stem, 54.0% showed cerebellar symptoms, 35.6% had pelvic organ involvement, and 55.6% experienced vision symptoms. There was a tendency for more frequent sensory symptoms among the female MS patients as compared with the males (*p* = 0.05) (Table 6). 

The *CXCL12* (rs1029153, rs1801157, and rs2297630) genotype distribution was not statistically significantly different in the CIS, RRMS, SPMS, and PPMS patient phenotype subgroups (Table 7). 

We analyzed whether there were associations between the prevalence of the affected functional systems and the *CXCL12* (rs1029153, rs1801157, and rs2297630) genotypes. None of the analyzed distributions reached the corrected significance level (*p* > 0.0167); however, there was a tendency for a higher prevalence of sensory system symptoms in the rs1801157 CC genotype patients compared with the CT genotype (*p* = 0.030, Table 8).

Since symptom prevalence was not homogeneously distributed between different genders, we analyzed the prevalence of affected functional systems after stratifying the subjects by gender. We found that the sensory symptoms were rarer in the women with the rs1801157 TT genotype when compared with the other genotypes (*p* = 0.004, Table 9). Additionally, in the men, there was a tendency, which almost reached statistical significance, of lower pyramidal symptom prevalence in the individuals with the rs1801157 TT genotype when compared with the other genotypes (*p* = 0.02). 

Finally, we analyzed the associations between the *CXCL12* (rs1029153, rs1801157, and rs2297630) genotypes and clinical parameters (age at MS diagnosis, MS duration, EDSS, and MSSS). Our analysis showed that there was a statistically significant association between a shorter MS disease duration and rs1801157 T/T genotype when compared with the other genotypes (*p* = 0.007), with the results shown in Table 10. 

## 3. Discussion

In this study, we investigated the *CXCL12* SNPs (rs1029153, rs1801157, rs2297630) genotypes and CXCL12 serum levels in the MS patients and control subjects. Our aim was to evaluate the associations between *CXCL12* SNPs, CXCL12 protein levels, MS prevalence, and clinical parameters. Our findings add to the expanding body of information about the genetic and molecular variables implicated in MS’s development and clinical course.

Our study showed that rs1801157 was associated with genetic risk only in male subjects. In the men, the CT genotype, when compared with the more common CC genotype, was associated with a 2.3 reduced odds of MS development, which possibly indicates a protective effect. A similar study by Azin et al. also reported an association between rs1801157 and MS. The researchers reported that the rs1801157 CT genotype was statistically significantly less prevalent in MS patients than in control individuals [22]. As of yet, *CXCL12* SNPs rs1029153 and rs2297630 have not been studied in MS patients. 

As previously mentioned, CXCL12 CSF levels and CXCL12 CNS histological changes positively correlate with disease severity, which indicates that CXCL12 could be used as a biomarker for MS [14,18], but there are limited data on CXCL12 serum levels. Azin et al. reported that serum CXCL12 levels were higher in MS patients than in control subjects [22]. We did not find any statistically significant differences in the CXCL12 serum levels between healthy subjects and MS patients. Edwards et al. studied CXCL12 levels in both CSF and serum, where the researchers found that while CXCL12 levels were significantly elevated in the CSF of MS patients, there were no significant differences in the serum [23]. Emamnejad et al. studied CXCL12 serum levels and peripheral blood mononuclear cell *CXCL12* gene expression in MS patients and healthy controls. They showed that *CXCL12* expression was increased in clinically active patients and decreased in patients who received IFNβ therapy when compared with healthy controls. Additionally, there were no significant differences in the CXCL12 serum levels between the studied groups [24]. In contrast, Azin et al. showed higher serum CXCL12 levels in MS patients compared with the control subjects. Also, the researchers did not find any associations between rs1801157 and the CXCL12 concentration [22]. Therefore, serum CXCL12 levels may not be as robust a biomarker for MS as its CSF levels or CNS histological changes.

We also studied the association between the CXCL12 serum levels and the *CXCL12* gene SNPs. Our data do not show any statistically significant associations between the rs1029153, rs1801157, and rs2297630 SNP genotypes and the CXCL12 concentration, with only a non-statistically significant tendency for a higher concentration in the rs1801157 CT genotype control group individuals when compared with the CC genotype subjects. As previously described, the rs1801157 CT genotype in the males was associated with a reduced risk of MS. Consequently, this genotype demonstrated two significant associations: a decreased risk of MS in the men and an elevated serum CXCL12 concentration in the control individuals. Studies showed that CXCL12 is important in adult neural progenitor cell chemotaxis after a CNS injury; constitutive expression also modulates neurotransmission and rescues neurons from apoptosis [25]. Animal studies also showed that blocking CXCL12 signaling significantly impairs remyelination in an MS-like model [26]. However, if CXCL12 expression is induced by inflammation, as is the case during MS, the chemokine may have a direct neurotoxic effect and attract infiltrating leukocytes, as described in the introduction. Therefore, CXCL12 may have a context-dependent effect: neuroprotective at rest, and detrimental if upregulated during inflammation. We hypothesize that the rs1801157 CT genotype could potentially increase the constitutive *CXCL12* gene expression, which results in an increased resting state serum concentration, and mitigates the early CNS damage, as well as promotes a more resilient neural environment. The mechanistic impact of the *CXCL12* rs1801157 SNP on gene expression—whether through alterations in the transcription factor binding, mRNA splicing, or other mechanisms—has not been described in the literature. However, it is known that some SNPs associated with multiple sclerosis (MS) can influence gene expression. Handel et al. studied the gene expression in a lymphoblastoid cell line derived from healthy volunteers. These volunteers were also genotyped for a substantial number of single-nucleotide polymorphisms (SNPs). The study revealed that alterations in gene expression were linked to 13 out of the 38 SNPs that were investigated [27]. Putscher et al. found that the SNPs rs3851808 (*EFCAB13*), rs1131123 (*HLA-C*), rs10783847 (*TSFM*), and rs2014886 (*TSFM*) may contribute to a differential splicing pattern, such as exon skipping in the *TSFM* gene SNP [28]. In silico studies showed that other MS-susceptibility-associated SNPs can impact miRNA binding and influence the overall expression [29]. As with most MS-associated SNPs [30], the *CXCL12* rs1801157 SNP is located in the untranslated region of the gene and could potentially affect regulatory element binding or mRNA splicing. However, further experimental studies are necessary to test this hypothesis. 

There is a growing body of evidence that shows various associations between MS clinical parameters and patient genotypes. We did not find differences in the rs1029153, rs1801157, and rs2297630 genotype distributions in different MS phenotypes, although the sample size for progressive disease phenotypes was small. As for affected functional systems, we found that sensory symptoms were rarer in women with the rs1801157 TT genotype when compared with the other genotypes. Furthermore, our data show lower pyramidal symptom prevalence among the male subjects with the rs1801157 TT genotype when compared with the other genotypes, although the results narrowly missed statistical significance, possibly due to the rarity of this genotype in our sample. Both functional system results could have been due to shorter disease duration in individuals with the rs1801157 TT genotype. To our knowledge, there were no studies that have reported associations between *CXCL12* gene SNPs, protein serum concentration, and clinical parameters. 

Our study had some limitations that should be considered when interpreting the findings. First, the sample size for the progressive disease phenotypes was small, which limited the statistical power. Second, due to multiple comparisons, some findings did not reach the adjusted statistical significance level, which indicates a need for a larger sample size. Additionally, we studied a single population in Lithuania, which could have limited the generalizability of our findings. Furthermore, the cross-sectional design of our study did not capture variations in the CXCL12 serum levels over the course of the disease and could only show associations but not prove causality. Clinical studies would be required to evaluate the significance of our findings. Finally, we did not account for some significant confounders, such as smoking, vitamin D levels, exercise, and treatment regimes. Most patients had a history of treatment with disease-modifying drugs, which further increased the heterogeneity of our sample. Further studies could account for these important confounders.

In conclusion, our study revealed some *CXCL12* SNP associations with MS risk, CXCL12 protein concentration, and clinical parameters. The rs1801157 CT genotype was associated with a reduced odds of MS in the men. In healthy individuals, this genotype was also associated with a higher CXCL12 serum concentration; however, the results did not reach adjusted statistical significance. Sensory systems were rarer in women with the rs1801157 TT genotype. These findings contribute to advancing our understanding of the genetic factors involved in MS pathogenesis. Further research is warranted to explore these associations and their importance in aiding predictions of the disease course and possibly the development of personalized therapeutic approaches. 

## 4. Materials and Methods

### 4.1. Ethical Statement and Study Participants

This study was conducted in accordance with the Declaration of Helsinki, and all participants (n = 500) gave written informed consent. The study protocol was approved by the Kaunas Regional Biomedical Research Ethics Committee (BE-2-/102). This study included two groups: MS group (n = 250): MS patients that received treatment at the Hospital of Lithuanian University of Health Sciences Kaunas Clinics from 1 January 2020 to 31 December 2023. The participants included individuals 18 years and older diagnosed with MS according to the revised 2017 McDonald criteria [16]. Disability was assessed using the Kurtzke Expanded Disability Status Scale (EDSS) before the venous blood draw, and was used for the MSSS calculation [31]. For the retrospective analysis, the following data were collected: age at the time of MS diagnosis; MS disease course and duration; presence of symptoms related to the pyramidal, sensory, brainstem, cerebellar, pelvic systems, and vision; and disability according the EDSS score. Control group (n = 250): Matched by sex and age and comprised individuals without medical history of autoimmune, oncological, or neurological diseases and in self-reported good health. Subjects with recent blood product transfusions, growth factor treatments, organ transplants, or diagnosed with oncological, systemic tissue, infectious, or other autoimmune diseases were excluded.


### 4.2. DNA Extraction and CXCL12 Genotyping

The DNA extraction and analysis of *CXCL12* rs1029153, rs1801157, and rs2297630 SNPs were conducted in the Ophthalmology Laboratory of the Neuroscience Institute of the Lithuanian University of Health Sciences. Peripheral venous blood was collected in vacuum tubes that contained ethylenediaminetetraacetic acid (EDTA) as an anticoagulant. The samples were subsequently stored at −70 °C until DNA extraction. DNA was isolated using the salting-out method. Briefly, the thawed blood samples were transferred to 15 mL centrifuge tubes and treated with lysis buffer I (8.3 g NH_4_Cl, 1 g KHCO_3_, 290.20 mg EDTA in 1 L of distilled water, pH = 7.4), followed by centrifugation at 1300× *g* for 15 min at 4 °C. The supernatant was discarded, and the lysis step was repeated three times. After the final wash, the leukocyte pellet was resuspended in 6 mL of lysis buffer II (1.58 g Tris HCl, 23.37 g NaCl, 584.40 mg EDTA in 1 L of distilled water, pH = 8.2), to which 150 μL of 10% sodium dodecyl sulphate (SDS) and 10 μL of proteinase K were added. The mixture was incubated at 56 °C for 10 min. After the incubation, 1 mL of 6 M NaCl was added, and the tubes were vigorously shaken. Following the addition of chloroform, the tubes were thoroughly mixed and centrifuged at 1500× *g* for 20 min at 16 °C to form a biphasic solution with a white precipitate of denatured proteins at the interface. The aqueous phase, which contained the DNA, was carefully transferred into new tubes, and an equal volume of cold 96% ethanol was added to precipitate the DNA. The DNA strands that formed were collected using a pipette tip and transferred into 1.5 mL tubes that contained 70% ethanol. These were centrifuged at 21,600× *g* for 3 min to pellet the DNA. The ethanol was carefully removed, and the residual ethanol was evaporated by placing the tubes in a 37 °C block. Finally, the DNA was resuspended in 100 μL of elution buffer and left in the refrigerator at 4 °C overnight to ensure complete dissolution. The DNA was stored at −20 °C until further analysis. The concentration and purity of the DNA were determined using an Agilent Technologies Cary 60 UV–Vis spectrophotometer (Agilent Technologies, Santa Clara, CA, USA).

The genotyping of *CXCL12* SNPs (rs1029153 (assay ID C___7550610_10), rs1801157 (assay ID C___3223115_10), rs2297630 (assay ID C___3223122_1_)) was performed using the StepOnePlus real-time PCR system (Applied Biosystems, Foster City, CA, USA). For each assay, 1.5 μL of DNA, 5 μL of TaqMan Universal Master Mix II (no UNG), 0.25 μL of primers and probes, and 3.25 μL of nuclease-free water were added to each well of a 96-well plate. The plate was sealed with an optical film, centrifuged briefly, and loaded into the thermocycler. Real-time PCR was conducted under the following cycling conditions: denaturation at 95 °C for 15 s, followed by hybridization and elongation at 60 °C for 60 s for a total of 45 cycles. Nuclease-free water was used as a negative control.

### 4.3. CXCL12 Protein Concentration Measurement

The CXCL12 serum concentrations for 40 control group and 40 MS group subjects were quantified using a Human CXCL12 ELISA Kit (Abbexa LTD, Cambridge, UK) with a detection range of 3.12–200.00 pg/mL. The samples were assayed following the manufacturer’s protocol. After the preparation and incubation of standards and test samples, the optical density was measured at 450 nm using a microplate reader. The concentrations were determined from a standard curve established with prepared CXCL12 solutions. For 2 control group individuals, the detected CXCL12 concentration fell outside the detection range; therefore, these data were excluded from the analysis.

### 4.4. Statistical Analysis

The MSSS values of the MS patients were calculated using the ms.sev package (Helga Westerlind et al., version 1.0.4) of R Studio (R version 4.3.3, R foundation for Statistical Computing, Austria) [32]. The MSSS is calculated in this package from the Global MSSS database, which includes 9892 patients from 11 countries [33]. The MSSS for each patient was calculated with the global_msss function based on the patient’s EDSS and disease duration.

Statistical data analysis was performed using the statistical program package “Statistical Package for the Social Sciences, version 29.0 for Windows” (SPSS for Windows, IBM Corp., Armonk, NY, USA). The distribution of the CXCL12 serum concentration values was assessed for normality using the Shapiro–Wilk test for sample groups with fewer than 50 individuals, and the Kolmogorov–Smirnov test for other quantitative variables. The data for these variables did not conform to a normal distribution. Therefore, descriptive statistics are reported using the median and interquartile range (IQR), specifically between the 25th and 75th percentiles. For comparisons between two groups of quantitative variables, the Mann–Whitney test was utilized. In cases that involved three or more groups, the Kruskal–Wallis test was employed. 

The genotype distributions of polymorphisms within the study groups were analyzed for conformity to the Hardy–Weinberg equilibrium model. Descriptive statistics for the qualitative criteria are expressed as percentages. The relationship between the frequencies of two characteristics was assessed using the chi-squared (χ^2^) criterion. If the subgroup sizes fell below five individuals, Fisher’s exact test was employed instead of the χ2 criterion. For the comparisons of paired data, the Z test with a Bonferroni correction was utilized. The odds ratio (OR) for the occurrence of MS was estimated through binary logistic regression while accounting for inheritance patterns and genotype combinations and reported with a 95% confidence interval (CI). The model accuracy was evaluated using the Akaike information criterion (AIC), where a lower value indicates a more suitable model. Statistical significance was established at a *p*-value of less than 0.05 and, where necessary, our significance threshold was adjusted for multiple comparisons to be less than 0.0167 (0.05/3) since three *CXCL12* SNPs were evaluated.

## Figures and Tables

**Table 1 ijms-25-09554-t001:** Characteristics of study participants.

Characteristic	Group		*p*-Value
	Control (n = 250)	MS (n = 250)	
Men n (%)	85 (34.0)	85 (34.0)	1.000
Women n (%)	165 (66.0)	165 (66.0)	
Men median age (IQR)	44 (27)	40 (14)	0.094
Women median age (IQR)	37 (24)	38 (16)	0.120

IQR—interquartile range, MS—multiple sclerosis.

**Table 2 ijms-25-09554-t002:** *CXCL12* (rs1029153, rs1801157, and rs2297630) genotype distributions in multiple sclerosis and control groups.

*CXCL12* SNP	Genotype/Allele	Control Group n (%)	Control Group HWE *p*-Value	MS Group n (%)	MS Group HWE *p*-Value	Difference across Groups *p*-Value
rs1029153	AA	124 (49.6)	0.903	132 (52.8)	0.186	0.591
	AG	102 (40.8)		91 (36.4)		
	GG	24 (9.6)		27 (10.8)		
	A allele	350 (70.0)		355 (71.0)		0.622
	G allele	150 (30.0)		145 (29.0)		
rs1801157	CC	152 (60.8)	0.824	166 (66.4)	0.348	0.244
	CT	88 (35.2)		71 (28.4)		
	TT	10 (4.0)		13 (5.2)		
	C allele	392 (78.4)		403 (80.6)		0.260
	T allele	108 (21.6)		97 (19.4)		
rs2297630	GG	129 (51.6)	0.659	139 (55.6)	0.508	0.659
	AG	97 (38.8)		90 (36.0)		
	AA	24 (9.6)		21 (8.4)		
	G allele	355 (71.0)		368 (73.6)		0.362
	A allele	145 (29.0)		132 (26.4)		

IQR—interquartile range, MS—multiple sclerosis, H-W—Hardy–Weinberg distribution, SNP—single-nucleotide polymorphism. Bonferroni-corrected significance level *p* < 0.0167.

**Table 3 ijms-25-09554-t003:** *CXCL12* (rs1029153, rs1801157, and rs2297630) genetic model binary logistic regression.

Model	Genotype/Allele	Odds Ratio (95% CI)	*p*-Value	AIC
*CXCL12* (rs1029153)				
Codominant	AG vs. AA	0.84 (0.58–1.22)	0.355	692.093
	GG vs. AA	1.06 (0.58–1.93)	0.857	
Dominant	AG+GG vs. AA	0.88 (0.62–1.25)	0.474	692.635
Recessive	GG vs. AA+AG	1.14 (0.64–2.04)	0.658	692.951
Overdominant	AG vs. GG+AA	0.83 (0.58–1.19)	0.312	692.126
Additive	G	0.96 (0.74–1.24)	0.738	693.035
*CXCL12* (rs1801157)				
Codominant	CT vs. CC	0.74 (0.5–1.08)	0.121	690.317
	TT vs. CC	1.19 (0.51–2.79)	0.689	
Dominant	CT+TT vs. CC	0.79 (0.55–1.13)	0.194	691.453
Recessive	TT vs. CC+CT	1.32 (0.57–3.06)	0.523	692.736
Overdominant	CT vs. TT+CC	0.73 (0.5–1.07)	0.103	690.478
Additive	T	0.88 (0.65–1.19)	0.395	692.422
*CXCL12* (rs2297630)				
Codominant	AG vs. GG	0.86 (0.59–1.25)	0.433	692.312
	AA vs. GG	0.81 (0.43–1.53)	0.519	
Dominant	AG+AA vs. GG	0.85 (0.6–1.21)	0.370	692.343
Recessive	AA vs. GG+AG	0.86 (0.47–1.6)	0.639	692.927
Overdominant	AG vs. AA+GG	0.89 (0.62–1.28)	0.518	692.729
Additive	A	0.89 (0.68–1.16)	0.374	692.355

CI—confidence interval, AIC—Akaike information criterion. Bonferroni-corrected significance level *p* < 0.0167.

**Table 4 ijms-25-09554-t004:** Gender-stratified binary logistic regression analysis of *CXCL12* (rs1029153, rs1801157, and rs2297630) genetic models.

Women				
Model	Genotype/Allele	Odds Ratio (95% CI)	*p*-Value	AIC
*CXCL12* (rs1029153)				
Codominant	AG vs. AA	0.66 (0.41–1.04)	0.075	452.270
	GG vs. AA	1.37 (0.65–2.86)	0.409	
Dominant	AG+GG vs. AA	0.77 (0.5–1.18)	0.226	456.006
Recessive	GG vs. AA+AG	1.66 (0.82–3.37)	0.161	455.467
Overdominant	AG vs. GG+AA	0.62 (0.4–0.97)	0.034	452.960
Additive	G	0.96 (0.7–1.33)	0.806	457.417
*CXCL12* (rs1801157)				
Codominant	CT vs. CC	0.99 (0.62–1.59)	0.964	457.176
	TT vs. CC	1.35 (0.45–4.01)	0.594	
Dominant	CT+TT vs. CC	1.03 (0.65–1.62)	0.908	457.464
Recessive	TT vs. CC+CT	1.35 (0.46–3.98)	0.586	457.178
Overdominant	CT vs. TT+CC	0.97 (0.61–1.56)	0.904	457.463
Additive	T	0 (0–0)	0.000	0.000
*CXCL12* (rs2297630)				
Codominant	AG vs. GG	0.71 (0.45–1.12)	0.139	454.959
	AA vs. GG	1.07 (0.49–2.3)	0.871	
Dominant	AG+AA vs. GG	0.77 (0.5–1.18)	0.226	456.009
Recessive	AA vs. GG+AG	1.24 (0.59–2.6)	0.572	457.156
Overdominant	AG vs. AA+GG	0.7 (0.45–1.09)	0.115	454.986
Additive	A	0.89 (0.64–1.24)	0.503	457.027
**Men**				
**Model**	**Genotype/Allele**	**Odds Ratio (95% CI)**	***p*-Value**	**AIC**
*CXCL12* (rs1029153)				
Codominant	AG vs. AA	1.42 (0.73–2.74)	0.301	232.733
	GG vs. AA	0.53 (0.17–1.67)	0.278	
Dominant	AG+GG vs. AA	1.16 (0.63–2.13)	0.641	235.452
Recessive	GG vs. AA+AG	0.47 (0.15–1.44)	0.184	233.810
Overdominant	AG vs. GG+AA	1.54 (0.81–2.92)	0.193	233.961
Additive	G	0.95 (0.6–1.5)	0.814	235.615
*CXCL12* (rs1801157)				
Codominant	CT vs. CC	**0.43 (0.22–0.83)**	**0.012**	**229.004**
	TT vs. CC	0.93 (0.24–3.66)	0.913	
Dominant	CT+TT vs. CC	0.48 (0.26–0.89)	0.020	230.147
Recessive	TT vs. CC+CT	1.27 (0.33–4.89)	0.732	235.552
Overdominant	CT vs. TT+CC	**0.43 (0.23–0.83)**	**0.011**	**229.016**
Additive	T	0.62 (0.37–1.05)	0.074	232.385
*CXCL12* (rs2297630)				
Codominant	AG vs. GG	1.31 (0.68–2.51)	0.424	232.142
	AA vs. GG	0.41 (0.12–1.39)	0.152	
Dominant	AG+AA vs. GG	1.05 (0.57–1.93)	0.876	235.646
Recessive	AA vs. GG+AG	0.37 (0.11–1.23)	0.105	232.783
Overdominant	AG vs. AA+GG	1.45 (0.76–2.75)	0.256	234.374
Additive	A	0.87 (0.54–1.39)	0.552	235.315

CI—confidence interval, AIC—Akaike information criterion. Bonferroni-corrected significance level *p* < 0.0167. Statistically significant results are shown in bold.

**Table 5 ijms-25-09554-t005:** CXCL12 serum protein concentration distribution across multiple sclerosis and control groups based on *CXCL12* (rs1029153, rs1801157, and rs2297630) genotypes.

*CXCL12* SNP	Genotype	Number of Individuals with Genotype n (%)		CXCL12 Concentration (pg/mL) Median (IQR)		CXCL12 Concentration Comparison across Groups (*p*-Value)
		Control (n = 38)	MS (n = 40)	Control	MS	
rs1029153	AA	22 (57.9)	20 (50.0)	37.72 (6.84)	36.575 (10.54)	0.314
	AG	10 (26.3)	12 (30.0)	37.55 (8.9)	33.75 (7.41)	0.254
	GG	6 (15.8)	8 (20.0)	35.815 (0.92)	34.59 (17.94)	0.985
Genotype distribution or CXCL12 concentration differences (*p*-value)		0.774		0.560	0.738	
rs1801157	CC	22 (57.9)	31 (77.5)	36.14 (4.79)	35.39 (10.19)	0.607
	CT	13 (34.2)	9 (22.5)	39.72 (5.45)	36.74 (17.09)	0.164
	TT	3 (7.9)	0 (0.0)	30.35 (15.4)	- *	-
Genotype distribution or CXCL12 concentration differences (*p*-value)		0.074		0.028	0.910	
rs2297630	GG	22 (57.9)	23 (57.5)	37.08 (5.93)	36.41 (10.26)	0.401
	GA	9 (23.7)	12 (30.0)	39.1 (7.94)	35.075 (6.71)	0.310
	AA	7 (18.4)	5 (12.5)	35.92 (4.81)	30 (18.41)	0.639
Genotype distribution or CXCL12 concentration differences (*p*-value)		0.693		0.678	0.976	

SNP—single-nucleotide polymorphism, MS—multiple sclerosis, IQR—interquartile range. Bonferroni-corrected significance level *p* < 0.0167. *—no individuals with this genotype.

**Table 6 ijms-25-09554-t006:** Multiple sclerosis group disease phenotypes and affected functional systems.

Variable	Gender		*p*-Value
	Women n (%)	Men n (%)	
CIS	5 (3.0)	5 (5.9)	0.194
RR	153 (92.7)	72 (84.7)	
SP	2 (1.2)	1 (1.2)	
PP	5 (3.0)	7 (8.2)	
Pyramidal system symptoms n (%)	141 (85.5)	77 (90.6)	0.250
Sensory system symptoms n (%)	112 (67.9)	47 (55.3)	0.050
Brainstem system symptoms n (%)	96 (58.2)	50 (58.8)	0.922
Cerebellar system symptoms n (%)	88 (53.3)	47 (55.3)	0.786
Pelvic system symptoms n (%)	59 (35.8)	30 (35.3)	0.942
Visual system symptoms n (%)	89 (53.9)	50 (58.8)	0.462

CIS—clinically isolated syndrome, RR—relapsing–remitting, SP—secondary progressive, PP—primary progressive.

**Table 7 ijms-25-09554-t007:** Analysis of multiple sclerosis phenotype and *CXCL12* (rs1029153, rs1801157, and rs2297630) genotype distribution.

*CXCL12* SNP	Genotype	Phenotype n (%)				*p*-Value
		CIS	RRMS	SPMS	PPMS	
rs1029153	AA	5 (50.0)	121 (53.8)	0 (0.0)	6 (50.0)	0.225
	AG	4 (40.0)	81 (36.0)	1 (33.3)	5 (41.7)	
	GG	1 (10.0)	23 (10.2)	2 (66.7)	1 (8.3)	
rs1801157	CC	8 (80.0)	147 (65.3)	3 (100.0)	8 (66.7)	0.461
	CT	2 (20.0)	67 (29.8)	0 (0.0)	2 (16.7)	
	TT	0 (0.0)	11 (4.9)	0 (0.0)	2 (16.7)	
rs2297630	GG	5 (50.0)	127 (56.4)	0 (0.0)	7 (58.3)	0.097
	AG	5 (50.0)	79 (35.1)	1 (33.3)	5 (41.7)	
	AA	0 (0.0)	19 (8.4)	3 (66.7)	0 (0.0)	

SNP—single-nucleotide polymorphism, CIS—clinically isolated syndrome, RR—relapsing–remitting, SP—secondary progressive, PP—primary progressive. Bonferroni-corrected significance level *p* < 0.0167.

**Table 8 ijms-25-09554-t008:** Analysis of affected functional systems and *CXCL12* (rs1029153, rs1801157, and rs2297630) genotype distribution.

*CXCL12* SNP	Genotype	Patients Affected by Symptoms in Different Systems n (%)					
		Pyramidal	Sensory	Brainstem	Cerebellar	Pelvic	Visual
rs1029153	AA	110 (83.3)	83 (62.9)	75 (56.8)	74 (56.1)	46 (34.8)	73 (55.3)
	AG	84 (92.3)	58 (63.7)	56 (61.5)	48 (52.7)	34 (37.4)	51 (56.0)
	GG	24 (88.9)	18 (66.7)	15 (55.6)	13 (48.1)	9 (33.3)	15 (55.6)
*p*-value		0.138	0.932	0.743	0.721	0.897	0.994
rs1801157	CC	145 (87.3)	114 (68.7)	99 (59.6)	91 (54.8)	60 (63.1)	93 (56.0)
	CT	64 (90.1)	40 (56.3)	40 (56.3)	36 (50.7)	27 (38.0)	39 (54.9)
	TT	9 (69.2)	5 (38.5)	7 (53.8)	8 (61.5)	2 (15.4)	7 (53.8)
*p*-value		0.116	0.030	0.844	0.722	0.283	0.98
rs2297630	GG	117 (84.2)	88 (63.3)	81 (58.3)	79 (56.8)	51 (36.7)	79 (56.8)
	AG	82 (91.1)	58 (64.4)	55 (61.1)	46 (51.1)	32 (35.6)	48 (53.3)
	AA	19 (90.5)	13 (61.9)	10 (47.6)	10 (47.6)	6 (28.6)	12 (57.1)
*p*-value		0.276	0.971	0.528	0.578	0.769	0.864

SNP—single-nucleotide polymorphism. Bonferroni-corrected significance level *p* < 0.0167.

**Table 9 ijms-25-09554-t009:** Gender-stratified analysis of affected functional systems and *CXCL12* (rs1029153, rs1801157, and rs2297630) genotype distribution.

Women							
*CXCL12* SNP	Genotype	Patients Affected by Symptoms in Different Systems n (%)					
		Pyramidal	Sensory	Brainstem	Cerebellar	Pelvic	Visual
rs1029153	AA	69 (82.1)	59 (70.2)	43 (51.2)	46 (54.8)	31 (36.9)	45 (53.6)
	AG	53 (89.8)	39 (66.1)	42 (71.2)	33 (55.9)	23 (39.0)	33 (55.9)
	GG	19 (86.4)	14 (63.6)	11 (50.0)	9 (40.9)	5 (22.7)	11 (50.9)
*p*-value		0.435	0.786	0.041	0.451	0.379	0.889
rs1801157	CC	93 (86.1)	**81 (75.0)**	66 (61.1)	60 (55.6)	42 (38.9)	59 (54.6)
	CT	42 (85.7)	**29 (59.2)**	26 (53.1)	23 (46.9)	17 (34.7)	27 (55.1)
	TT	6 (75.0)	**2 (25.0)**	4 (50.0)	5 (62.5)	0 (0.0)	3 (37.5)
*p*-value		0.69	**0.004**	0.569	0.525	0.085	0.632
rs2297630	GG	75 (83.3)	63 (70.0)	48 (53.3)	50 (55.6)	35 (38.9)	50 (55.6)
	AG	51 (87.9)	39 (67.2)	41 (70.7)	31 (53.4)	21 (36.2)	30 (51.7)
	AA	15 (88.2)	10 (58.8)	7 (41.2)	7 (41.2)	3 (17.6)	9 (52.9)
*p*-value		0.698	0.658	0.037	0.552	0.245	0.898
**Men**							
***CXCL12* SNP**	**Genotype**	**Patients Affected by Symptoms in Different Systems n (%)**					
		**Pyramidal**	**Sensory**	**Brainstem**	**Cerebellar**	**Pelvic**	**Visual**
rs1029153	AA	41 (85.4)	24 (50.0)	32 (66.7)	28 (58.3)	15 (31.3)	28 (58.3)
	AG	31 (96.9)	19 (59.4)	14 (43.8)	15 (46.9)	11 (34.4)	18 (56.3)
	GG	5 (100.0)	4 (80.0)	4 (80.0)	4 (80.0)	4 (80.0)	4 (80.0)
*p*-value		0.173	0.369	0.076	0.312	0.094	0.601
rs1801157	CC	52 (89.7)	33 (56.9)	33 (56.9)	31 (53.4)	18 (31.0)	34 (58.6)
	CT	22 (100.0)	11 (50.0)	14 (63.6)	13 (59.1)	10 (45.5)	12 (54.5)
	TT	3 (60.0)	3 (60.0)	3 (60.0)	3 (60.0)	2 (40.0)	4 (80.0)
*p*-value		0.02	0.838	0.86	0.881	0.471	0.579
rs2297630	GG	42 (85.7)	25 (51.0)	33 (67.3)	29 (59.2)	16 (32.7)	29 (59.2)
	AG	31 (96.9)	19 (59.4)	14 (43.8)	15 (46.9)	11 (34.4)	18 (56.3)
	AA	4 (100.0)	3 (75.0)	3 (75.0)	3 (75.0)	3 (75.0)	3 (75.0)
*p*-value		0.196	0.547	0.086	0.397	0.232	0.77

SNP—single-nucleotide polymorphism. Bonferroni-corrected significance level *p* < 0.0167. Statistically significant results are shown in bold.

**Table 10 ijms-25-09554-t010:** Analysis of disease characteristics and *CXCL12* (rs1029153, rs1801157, and rs2297630) genotype distribution.

*CXCL12* SNP	Genotype	Age of MS Diagnosis Median (IQR)	MS Duration Years Median (IQR)	EDSS Median (IQR)	MSSS Median (IQR)
rs1029153	AA	29 (15)	5 (8)	3 (2)	5.93 (2.3)
	AG	32 (17)	8 (7)	3.5 (2)	5.57 (2.1)
	GG	33 (13)	5 (7)	3.5 (2)	6.46 (1.9)
*p*-value		0.085	0.078	0.660	0.335
rs1801157	CC	32 (16)	**6 (7)**	3.5 (2.5)	5.995 (2.2)
	CT	30 (13)	**7 (9)**	3.5 (2)	5.46 (2.4)
	TT	39 (15.5)	**3 (2)**	2.5 (0.3)	6.37 (1.7)
*p*-value		0.262	**0.007**	0.261	0.677
rs2297630	GG	29 (15)	5 (8)	3 (2)	5.93 (2.3)
	AG	32 (16.3)	7.5 (7)	3.5 (2.1)	5.515 (1.9)
	AA	35 (8)	5 (7)	3.5 (2)	6.46 (1.9)
*p*-value		0.103	0.335	0.851	0.299

SNP—single-nucleotide polymorphism, MS—multiple sclerosis, EDSS—Expanded Disability Status Scale, MSSS—Multiple Sclerosis Severity Score, IQR—interquartile range. Bonferroni-corrected significance level *p* < 0.0167. Statistically significant results are shown in bold.

## Data Availability

All data relevant to the study are included in this article.

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
