# Peer review of "CXCL12 Gene Polymorphisms and Serum Levels: Associations with Multiple Sclerosis Prevalence and Clinical Parameters in Lithuania"

_ijms, 2024, doi:10.3390/ijms25179554_

Round 1

Reviewer 1 Report (New Reviewer)

Comments and Suggestions for Authors

Dear Authors, I reviewed with pleasure the paper entitled “CXCL12 Gene Polymorphisms and Serum Levels: Associations 2 with Multiple Sclerosis Prevalence and Clinical Parameters in 3 Lithuania”

The paper is well-written, and the topic is of interest.

The introduction is clear and relevant, and the methods are exhaustive. The results and discussion are clear and of interest.

I just have a curiosity: Were all patients drug-naive? If so, I suggest highlighting this data. If not, I encourage the Authors to address these issues that could potentially limit some results.

Thank you 

Comments on the Quality of English Language

english is fine

Author Response

Dear Reviewer,

Thank you for the thoughtful and encouraging review of our manuscript. We appreciate your positive feedback.

As for the question on MS therapy. The patients in our study were mostly not drug naive. We have added some clarification in our Discussion section, pointing out this as a limitation. We hope it adresses your concerns.

Thank you.

Reviewer 2 Report (New Reviewer)

Comments and Suggestions for Authors

Valiukevicius et al. conducted a comprehensive study investigating the association between genetic variations in the CXCL12 gene, its corresponding protein levels, and the development and progression of multiple sclerosis (MS). Their findings revealed that a specific genetic variant (rs1801157 CT) in men was linked to a reduced risk of MS, while in women, another variant (rs1801157 TT) was associated with fewer sensory symptoms and a shorter disease course.

One minor comment: Authors' data presentation is commendable; I would recommend adding a few lines to the conclusion emphasizing the significance of this study in advancing our understanding of MS pathogenesis.

Author Response

Dear Reviewer,

Thank you for reviewing our manuscript. We appreciate your time and effort. 

We have expanded the conclusion section with some of our thoughts on how these findings are relevant to furthering our understanding of MS and how, hopefully, they could be used to better diagnose and treat MS. We hope to have fully adressed your concerns. 

Thank you.

This manuscript is a resubmission of an earlier submission. The following is a list of the peer review reports and author responses from that submission.

Round 1

Reviewer 1 Report

Comments and Suggestions for Authors

The manuscript entitled “CXCL12 Gene Polymorphisms and Serum Levels: Associations with Multiple Sclerosis Prevalence and Clinical Parameters in Lithuania” deals with the involvement of CXCL12 variants in the genetic basis of MS. The authors selected three potentially functional variants for their analysis of CXCL12-related MS susceptibility and further analysed the relation of selected SNPs with the disease characteristics and with the serum level of CXCL12.  The topic of the manuscript is within the scope of the journal and would be interesting to readers involved in research on the molecular pathogenesis and genetics of MS and the related autoimmune disorders. However, there are some serious concerns and issues that need to be addressed and some corrections are required (in order of appearance):

- There are various grammar errors throughout the manuscript. For instance, line 17, “with diagnosed” should be “diagnosed with”.

- SNP IDs should be deleted from line 20 in the Abstract section.

- Line 18: “SNPs were genotyped with real-time PCR” should be “SNPs were genotyped with real-time PCR-based assays”.

- The Introduction should include the explanation why these exact SNPs were selected. I suggest that the authors rephrase and include section 4.2 in the Introduction. Furthermore, potential functional consequences of the presence of variant alleles should be stated (Introduction or the Discussion section), like introducing/changing/destroying microRNA binding sites, or affecting mRNA splicing, etc.

- Selecting the best-fitting model according to AIC score is inappropriate when a difference between score acquired for the compared models is less than 2. Therefore, the statement in lines 146 and 147 should be rephrased.

- Table 10 should include numbers of individuals with specific genotypes in order to be more informative.

- There are too many Tables for this type of the article (16). Some should be transferred to Supplement in order to make the article more concise and easier to read.

- Any correlation coefficient less than 0.3 indicated the lack of correlation. Therefore, the associations between alleles and the age of MS onset are negligible and biologically and medically irrelevant.  Therefore, corrections are needed in the Results section, as well as in the Discussion and Abstract. Additionally, criteria for the interpretation of the results of correlation test should be corrected (lines 443 and 444).

- Some additional information about the recruitment of controls are needed: where and how were these individuals recruited? Were they self-reported as “in good health” or they passed some physical examination?   

- For all buffers used for sample preparation and DNA extraction the composition should be stated, together with pH, where necessary. Centrifugal force should be stated as RCF (g), not as rpm, since it should be universal , not dependent on the exact centrifuge used for the specific experiment.

- The exact names and IDs of the assays used for genotyping should be stated.

Comments on the Quality of English Language

Miner editing is needed.

Author Response

Dear Reviewer,

            Thank you for the thorough and valuable feedback. We have considered your suggestions and made the necessary revisions to address your concerns. We will re-upload the revised manuscript with tracked changes. Below please find out point-by-point response to the comments.

Comment (C): There are various grammar errors throughout the manuscript. For instance, line 17, “with diagnosed” should be “diagnosed with”. 

Answer (A): Dear reviewer, thank you for your comment; we have revised the manuscript for grammar errors.

C: SNP IDs should be deleted from line 20 in the Abstract section.

A: Dear reviewer, thank you, it was deleted.

C: Line 18: “SNPs were genotyped with real-time PCR” should be “SNPs were genotyped with real-time PCR-based assays”.

A: Dear reviewer, thank you, it was corrected.

C: The Introduction should include the explanation why these exact SNPs were selected. I suggest that the authors rephrase and include section 4.2 in the Introduction. Furthermore, potential functional consequences of the presence of variant alleles should be stated (Introduction or the Discussion section), like introducing/changing/destroying microRNA binding sites, or affecting mRNA splicing, etc.

A: Dear reviewer, thank you, it was revised.

C: Selecting the best-fitting model according to AIC score is inappropriate when a difference between score acquired for the compared models is less than 2. Therefore, the statement in lines 146 and 147 should be rephrased.

A: Dear reviewer, thank you, it was corrected.

C: Table 10 should include numbers of individuals with specific genotypes in order to be more informative.

A: Dear reviewer, thank you, the information was added.

C: There are too many Tables for this type of the article (16). Some should be transferred to Supplement in order to make the article more concise and easier to read.

A: Dear reviewer, thank you for the suggestion; we have created a Supplemetary material file according to your suggestion.

C: Any correlation coefficient less than 0.3 indicated the lack of correlation. Therefore, the associations between alleles and the age of MS onset are negligible and biologically and medically irrelevant.  Therefore, corrections are needed in the Results section, as well as in the Discussion and Abstract. Additionally, criteria for the interpretation of the results of correlation test should be corrected (lines 443 and 444).

A: Dear reviewer, thank you for your feedback regarding the interpretation of the correlation coefficients in our manuscript. We appreciate the opportunity to clarify our position.

Regarding the comment on the correlation coefficient, we respectfully disagree with the assertion that a coefficient less than 0.3 indicates a complete lack of correlation. While it is true that a coefficient in this range indicates a weak correlation, this does not imply that the correlation is irrelevant or biologically insignificant. In many research contexts, particularly in complex fields like the study of MS, even weak correlations can provide valuable insights and contribute to our understanding of the disease mechanisms.

In our study, the weak correlations observed between certain alleles and the age of MS onset were statistically significant. Although the correlations are not strong, they are present and could be noteworthy within the specific context of MS research. These findings may warrant further investigation and could have implications for understanding the genetic factors influencing MS onset.

C: Some additional information about the recruitment of controls are needed: where and how were these individuals recruited? Were they self-reported as “in good health” or they passed some physical examination?  

A: Dear reviewer, we have modified the description in the methods section to include this information.

C: For all buffers used for sample preparation and DNA extraction the composition should be stated, together with pH, where necessary. Centrifugal force should be stated as RCF (g), not as rpm, since it should be universal, not dependent on the exact centrifuge used for the specific experiment.

A: Dear reviewer, thank you for your comment, the information was added to the main manuscript.

C: The exact names and IDs of the assays used for genotyping should be stated.

A: Dear reviewer, please find below the requested information that was added to the manuscript:

Rs1029153 assay ID C___7550610_10;

Rs1801157 assay ID C___3223115_10;

Rs2297630 assay ID C___3223122_1;

We hope these revisions meet your expectations. Thank you for your time and valuable constructive feedback.

Reviewer 2 Report

Comments and Suggestions for Authors

The authors cite the report that previously identified 233 MS-risk polymorphisms. The mechanisms underlying those GWAS associations are mostly unknown; therefore, a wide research field is open.

Trying to prove additional candidate polymorphisms in MS using low-powered cohorts reproduces the mistake of a pre-GWAS era. Considering that some clinical data is collected, it would have been more interesting to focus on any of those 233 variants and delve into their functional impact on the disease. As it is, no comparison stands Bonferroni correction for the multiple comparisons performed, and most probably the significances shown are spurious associations. 

Author Response

Please see the attachement for our reply.

Round 2

Reviewer 1 Report

Comments and Suggestions for Authors

The authors have made certain corrections that significantly improved their manuscript. I disagree with their explanation for the results of correlation analysis. The number of groups according to the allele carrier status is just three (0, 1 and 2) which in combination with very low correlation coefficients results in a negligible effect on the age of disease onset, for instance. The only reason for reaching statistical significance is the number of participants. Since the authors specifically highlighted the supposed correlations and included these results in the Abstract, the correction is required and caution is necessary when drawing conclusion based on their findings. 

Author Response

Dear Reviewer,

Thank you for the insightful comment. We have changed the abstract, results and conclusions to emphasize the weak and possibly clinically irrelevant correlation. We believe these modifications adress your concerns and provide a clearer interpretation of our findings.

Thank you for your time and feedback.